# Advancing Epidemiology and Genetic Approaches for the Treatment of Spinal and Bulbar Muscular Atrophy: Focus on Prevalence in the Indigenous Population of Western Canada

**DOI:** 10.3390/genes14081634

**Published:** 2023-08-17

**Authors:** Harry Wilton-Clark, Ammar Al-aghbari, Jessica Yang, Toshifumi Yokota

**Affiliations:** 1Department of Medical Genetics, Faculty of Medicine and Dentistry, University of Alberta, Edmonton, AB T6G 2R3, Canada; hwiltonc@ualberta.ca; 2Department of Biological Sciences, Faculty of Science, University of Alberta, Edmonton, AB T6G 2R3, Canada; aalaghba@ualberta.ca; 3Department of Immunology, Department of Pharmacology and Toxicology, Faculty of Arts and Science, University of Toronto, Toronto, ON M5S 1A1, Canada; jessicas.yang@mail.utoronto.ca

**Keywords:** spinal and bulbar muscular atrophy, antisense therapy, oligonucleotide, splice switching, mRNA knockdown, androgen receptor, AR45

## Abstract

Spinal and bulbar muscular atrophy (SBMA), also known as Kennedy’s disease, is a debilitating neuromuscular disease characterized by progressive muscular weakness and neuronal degeneration, affecting 1–2 individuals per 100,000 globally. While SBMA is relatively rare, recent studies have shown a significantly higher prevalence of the disease among the indigenous population of Western Canada compared to the general population. The disease is caused by a pathogenic expansion of polyglutamine residues in the androgen receptor protein, which acts as a key transcriptional regulator for numerous genes. SBMA has no cure, and current treatments are primarily supportive and focused on symptom management. Recently, a form of precision medicine known as antisense therapy has gained traction as a promising therapeutic option for numerous neuromuscular diseases. Antisense therapy uses small synthetic oligonucleotides to confer therapeutic benefit by acting on pathogenic mRNA molecules, serving to either degrade pathogenic mRNA transcripts or helping to modulate splicing. Recent studies have explored the suitability of antisense therapy for the treatment of SBMA, primarily focused on gene therapy and antisense-mediated mRNA knockdown approaches. Advancements in understanding the pathogenesis of SBMA and the development of targeted therapies offer hope for improved quality of life for individuals affected by this debilitating condition. Continued research is essential to optimize these genetic approaches, ensuring their safety and efficacy.

## 1. Introduction

### 1.1. Clinical Features and Prevalence in the Indigenous Population of Western Canada

Spinal and bulbar muscular atrophy (SBMA), also known as Kennedy’s disease, is an X-linked recessive neuromuscular disease with slow progression, caused by mutations in the androgen receptor gene and primarily characterized by degeneration of lower motor neurons [1,2]. The disease typically only affects males, but females can be carriers and sometimes experience muscle cramps. It typically has an onset at around 30–40 years of age in patients, with symptoms such as hand tremors, muscle cramps, and back pain arising first [1,3]. As the disease progresses, patients begin to experience muscle weakness, usually starting in the lower limbs. Within a few years after the onset of muscle weakness, many patients will require a handrail to climb stairs [3]. Fasciculations also appear early in the disease, commonly appearing in the lower areas of the face. As the bulbar muscles start to weaken, SBMA patients notice difficulties with speech and swallowing [2]. Large natural history studies have identified that these bulbar symptoms begin to appear at an average age of 50 [3]. Near the end of the disease, patients begin needing assistance from canes or wheelchairs. A common cause of death in SBMA patients is aspiration pneumonia due to bulbar weakness [3].

While primarily viewed as a lower motor neuron disorder, growing evidence has identified that peripheral muscle tissues also play an important role in the SBMA phenotype. Patients with SBMA have been found to have serum creatine kinase levels consistent with a myopathic cause, and skeletal muscle satellite cells from SBMA patients have been demonstrated to have reduced proliferative ability in vitro [4,5,6]. Furthermore, studies treating only the peripheral muscle and not the motor neurons of SBMA mice have still displayed notable improvement in the SBMA phenotype [7]. These findings suggest that at least some of the neuromuscular symptoms of the SBMA phenotype could be caused by SBMA-related myopathy rather than the neuropathy that was previously suspected.

Beyond the neuromuscular symptoms, SBMA also has an effect on endocrine function. Changes in the functionality of the androgen receptor result in partial androgen insensitivity in many patients, which often appears prior to any muscular symptoms [8,9,10]. Gynecomastia is the most common manifestation of androgen insensitivity in SBMA, with over 70% of patients experiencing it, according to multiple studies [8,9,10]. Other common symptoms include erectile dysfunction, testicular atrophy, and fertility problems. SBMA also has an effect on metabolism, and many patients display above-average levels of cholesterol and triglycerides, as well as elevated rates of diabetes, insulin resistance, and non-alcoholic fatty liver disease [10,11]. No cure currently exists for SBMA [12]. Typical management strategies include occupational and speech therapies, screening for respiratory problems, and pharmacological therapies to manage symptoms [13]. However, these approaches fail to address the underlying disease.

SBMA has a prevalence of 1–2 individuals per 100,000 people [2,14]. The mutation causing SBMA appears to have occurred independently in various populations worldwide, with different founder haplotypes of the disease found in Scandinavian countries, Japan, Germany, Italy, Australia, and Canada [15]. A higher prevalence of the disease in certain areas of the world is associated with founder effects. For example, in the region of Vaasa, Finland, the prevalence of SBMA is estimated to be around 7.65 per 100,000 [16,17]. This high prevalence has been attributed to a founder effect resulting from an ancient mutation in Western Finland [17]. A similar founder effect was also observed in a group of Japanese SBMA patients through genetic analysis [18]. 

The highest prevalence of SBMA in a population found thus far is in the Indigenous population of Saskatchewan, Canada, where the estimated prevalence is 14.7 per 100,000 people [16]. The authors of the study, Leckie et al., also noticed a high representation of patients with Saulteaux background (either from the clinic they were studying or reported relatives) and predicted an even higher prevalence of 184.24 per 100,000 people in the Saulteaux community of Saskatchewan [16]. Notably, two participants in the study were recruited from the neighboring province of Alberta, indicating that this high prevalence may extend beyond Saskatchewan [16]. Despite already demonstrating a much higher SBMA prevalence than any other population, the authors of this paper believe that these numbers could still be underestimated, as many individuals living with SBMA in the Indigenous communities do not currently go to the neuromuscular clinic [16]. Their findings suggest that the prevalence in Indigenous populations in Saskatchewan may represent the highest carrier rate for SBMA worldwide.

The high prevalence in the Indigenous communities of Saskatchewan has been associated with a founder effect, with the majority of patients in Leckie et al.’s study sharing the same haplotype [16]. Their findings suggest the founder effect likely originated approximately 250 years ago in this population. A second, distinct haplotype found in the study may also point to further founder effects in Métis and other Indigenous populations [16]. However, the authors mention that further research incorporating relatedness analyses, comprehensive genetic investigations, and larger ethnically matched control groups is necessary to strengthen these findings and gain more insights into the origin of the mutations.

Leckie et al. also explored whether Indigenous individuals with SBMA have different phenotypes compared to those documented previously in other parts of the world [16]. Like previous findings, their study indicates a slow progression of the disease, with the accumulation of weakness over time, as well as an inverse correlation between age at onset of SBMA and the size of the expanded androgen receptor (AR) repeat [16]. However, future studies with a larger sample size and more extended observation of the patients over time are needed to confirm potential phenotype differences in Indigenous populations.

The study also highlights the significant health disparities that Indigenous individuals still face in Canada, stemming from Canada’s colonial history and the wide array of social, economic, and political barriers present for Indigenous Canadians [16,19]. Indigenous populations often face barriers to accessing specialists for diagnosis and treatment in Canada, and are often not included in genetic research [20]. Recently, more care has been taken to involve Indigenous populations in genetic research whilst respecting their concerns, allowing better representation and the potential for improved future treatment of genetic conditions in Indigenous communities [16,21]. In terms of SBMA, Leckie et al. suggest that a Canadian disease registry for SBMA may help further research into this disease and guide clinical efforts toward critical areas [16].

### 1.2. Mechanism and Genetics

Spinal and bulbar muscular atrophy results from mutations in the *AR* gene encoding for the androgen receptor. The androgen receptor protein is one of many steroid hormone receptors belonging to the nuclear receptor superfamily and works to facilitate the effect of androgens on the body [22,23]. First discovered in humans in 1988, the *AR* gene is located on the long arm of the X chromosome at Xq11–12 and encodes for a 110 kDa protein product [24,25,26,27,28]. The *AR* gene is composed of eight exons that encode the three major domains of AR protein known as the transactivation domain, the DNA-binding domain, and the ligand-binding domain [22,29]. Interestingly, these three functional domains demonstrate differing degrees of conservation as the transactivation domain, coded by exon 1, is highly variable and known to have repeated glutamine, proline, and glycine residues [24]. In contrast, the DNA-binding domain encoded by exons 2 and 3 of the *AR* gene is conserved in all members of the steroid receptor superfamily [24,30]. The remaining exons 4–8 are less conserved and code for the ligand-binding domain of the *AR* gene [24,31]. 

SBMA belongs to a class of diseases known as trinucleotide repeat expansion disorders and results from a pathologic polyglutamine expansion in the first exon of the androgen receptor gene, characterized by glutamine residues encoded at a higher-than-normal range near the transcriptional activation domain of the AR protein [32]. In a healthy individual, there are typically between 9 and 36 CAG repeats in this region, and SBMA begins to manifest with more than 38 repeats present [33]. Although it is not the most well-known of the group, SBMA holds the title of the first identified repeat expansion disease. In addition to SBMA, eight other diseases stem from CAG-polyglutamine repeat expansions, all of which are characterized by their progressive neurodegenerative effects on humans [34]. These include Huntington’s disease, dentatorubral–pallidoluysian atrophy (DRPLA), and Spinocerebellar ataxia types 1, 2, 3, 6, 7, and 17 [34].

The androgen receptor functions as a ligand-dependent transcription factor that regulates, in coordination with other co-regulatory proteins, the transcription of the androgen-responsive genes by binding to their regulatory sites when in its androgen-bound form [22]. In its inactive form, AR is localized to the cytoplasm where it, like other members of the steroid hormone receptors family, forms a complex with heat shock proteins [35]. Upon androgen (testosterone or dihydrotestosterone) binding, AR undergoes a conformational change and a subsequent detachment of heat shock proteins, preparing AR for translocation into the nucleus [32,35,36,37,38,39,40,41]. Following androgen binding, AR undergoes dimerization and is targeted to the nucleus, where it binds to androgen-responsive elements found on androgen-regulated genes, regulating their expression [34].

SBMA is a disease characterized by both gain- and loss-of-function mechanisms [42] (Figure 1). The neurotoxic effects seen across the majority of polyglutamine expansion diseases indicate a toxic gain-of-function as the cause of neurodegeneration, consistent with the dominant pattern of inheritance in almost all these diseases [43,44]. This suggests that the encoded polyQ-AR protein is structurally modified in a way that enables it to introduce a new function that is pathogenic to neurons [43]. Interestingly, a secondary loss-of-function component is also believed to contribute to the pathogenesis of polyglutamine diseases, which manifests as partial androgen insensitivity [42]. 

The mechanism underlying the neurodegenerative gain-of-function symptoms present in SBMA is not yet fully understood [45]. A leading hypothesis is that the well-documented accumulation and aggregation of polyQ-AR in the nucleus could be contributing to disease and motor neuron degeneration. The formation of nuclear aggregates of mutant AR is evident in both motor neurons of the spinal cord and the brain stem as well as some non-neural tissue [46]. The expansion of the polyglutamine tract in AR affects the folding of the final AR product and is associated with an increase in α-helical structures [47,48,49]. Further, it increases the stability of α-helices via the unconventional hydrogen binding of the glutamine side chain and main chain carbonyl group [47,49]. It has also been reported that this abnormal hydrogen bonding plays a role in the formation of antiparallel β-strands of polyglutamine repeats into sheets or barrels [50]. These structural changes may result in abnormal protein–protein interaction and/or the subsequent degradation of mutant proteins [43]. Others have argued that aggregation arises as the expanded polyglutamine tract could serve as a substrate for the catalysis of cross-linked protein products via transglutaminase activity, leading to the formation of aggregates and their possible breakdown [43,44,51]. While it remains unclear exactly how these protein aggregates cause disease, they are thought to be closely related to the impaired axonal transport and ultimate neuronal degeneration observed in SBMA [52,53,54]. 

The loss-of-function mechanism has been better characterized and is a combination of androgen insensitivity and reduced trophic support. Androgen insensitivity arises due to a reduction in the appropriate function of polyQ-AR protein, corresponding with a decrease in the expression of genes typically activated by testosterone and dihydrotestosterone [31]. Additionally, polyQ-AR has been shown to display altered binding to CREB binding protein (CBP), leading to reduced transcriptional activity for genes modulated by this complex, most notably vascular endothelial growth factor (VEGF) [55]. VEGF has previously been reported to be downregulated in SBMA mice, and it is thought that this loss of trophic support could also contribute to the motor neuron pathogenesis of SBMA [42,55].

## 2. Current Antisense Approaches for SBMA

Antisense oligonucleotide therapy is a promising new area of drug development, and multiple drugs using this technology have received FDA approval in recent years [56]. This form of therapy uses oligonucleotides—often referred to as AONs or ASOs—to bind via Watson–Crick base pairing to a target sequence of RNA, where it interferes with gene expression [56,57]. One way this interference can occur is through the RNase H1 enzyme, which cleaves the targeted RNA, resulting in RNA degradation [58,59] (Figure 2). Another mechanism commonly used is splicing alteration, in which the oligonucleotide interferes with the normal splicing patterns of the target RNA, resulting in changes in exon skipping or inclusion [60]. Here, we aim to provide an overview of current antisense-based approaches to treat SBMA.

### 2.1. ASO-Mediated Androgen Receptor Knockdown

The purpose of antisense oligonucleotide (ASO)-mediated AR knockdown is to degrade mutant AR transcripts which are known to accumulate in the nucleus [54] (Figure 2). By targeting ASOs containing a ribonuclease-inducing core to the *AR* gene, the pathogenic AR mRNA and protein levels can be reduced, leading to an improved phenotype.

Lieberman et al. aimed to explore the effect of peripheral polyQ-AR suppression in transgenic SBMA mouse models [7]. Specifically, they focused on whether ASO-mediated knockdown of polyQ-AR could improve the peripheral muscular pathology of SBMA in these mice. The authors first identified two different 2′,4′-constrained ethyl (cEt) gapmers which caused a dose-dependent reduction in the AR mRNA of HUVEC cells, denoted ASO1 and ASO2. ASO1 targets a region in the AR transcript conserved between human and murine transcripts, whereas ASO2 targets a region specific to humans [7]. ASO1 and ASO2 were further validated in vivo in the AR113Q or humanized BAC fxAR121 SBMA mouse models, respectively. Consistent with their in vitro findings, the authors identified that subcutaneous ASO treatment led to a significant knockdown of AR mRNA and a nearly complete reduction (~90% for ASO1, 95% for ASO2) in AR protein levels in the mouse quadriceps muscle. This was accompanied by a clear improvement in certain aspects of the disease phenotype, with significant amelioration of grip strength, lean body mass, muscle fiber size, and lifespan in mice treated with either ASO1 or ASO2. Notably, AR expression was unaffected in the spinal cord following subcutaneous treatment, as gapmers with this chemistry are unable to permeate the blood–brain barrier [7]. This study served as preliminary justification for the therapeutic usage of ASOs to treat SBMA, and also demonstrated the important role that skeletal muscle plays in the SBMA phenotype.

In a following study, Sahashi et al. explored ASO-mediated knockdown of AR transcripts in SBMA mice but focused on neuronal polyQ-AR expression rather than peripheral [61]. The SBMA mouse model AR-97Q, which expresses both murine and transgenic human AR protein, was treated with intracerebroventricular (ICV) injection of either ASO-AR1 or ASO-2, both of which are 2′-MOE gapmers. ASO-AR1 used the same sequence as Liberman et al.’s ASO1 targeting both human and murine AR, while ASO-AR2 was a distinct sequence specific to mice [7,61]. Treatment with either ASO led to a ~50% decrease in mutant AR mRNA and protein in the spinal cord and brain, and ASO-AR1 showed an additional ~90% reduction in murine AR mRNA. AR levels in peripheral muscle were unaffected. Mice treated with either ASO also showed marked improvement in clinical phenotype, demonstrating significantly improved survival, grip strength, and rotarod performance compared to controls. Immunohistochemical analysis of ASO-AR1-treated mice also demonstrated numerous markers of improvement, such as reduced motor neuron shrinkage, reduced neuronal degeneration, and improved neuromuscular junction endplate maturation. Furthermore, despite negligible uptake of ASO into the skeletal muscle following ICV injection, the muscles of ASO-AR1-treated mice showed restored fiber size and reduced atrophy compared to controls [61].

Finally, Evers et al. aimed to identify a generic ASO candidate which could knock down transcripts from all diseases arising from CAG repeats, such as Huntington’s disease, SBMA, and the spinocerebellar ataxias [62]. Despite showing some benefit for Huntington’s disease, their ASO candidate failed to induce notable knockdown of AR mRNA when tested in vitro and is therefore unlikely to be applicable for SBMA [62]. Beyond the scope of SBMA, several groups have also explored antisense-mediated knockdown of AR to treat prostate cancer, with dozens of ASOs screened and several successful candidates identified [63,64]. While these ASOs have not been explored in the context of SBMA, they further confirm the ability of ASOs to knock down AR transcripts both in vitro and in vivo.

Taken together, these studies demonstrate that antisense-mediated knockdown of AR is highly effective. Furthermore, the findings of Lieberman et al. and Sahashi et al. demonstrate that AR knockdown in both peripheral tissues and the CNS is associated with an improved clinical phenotype for SBMA, suggesting that ASOs may have therapeutic potential for this indication [7,61]. Of note, both groups were limited by the inability of their gapmers to cross the blood–brain barrier, requiring mutually exclusive treatment of either the CNS or peripheral muscles based on injection type. Interestingly, recent studies have identified numerous nanocarriers and apoptotic bodies which can successfully facilitate oligonucleotide penetration of the blood–brain barrier [65,66,67]. While not yet explored for SBMA, these carriers could potentially enable the treatment of both CNS and peripheral tissues with a single injection. Given that SBMA is a disease with both CNS and peripheral muscle involvement, this could be a major next step in improving the therapeutic efficacy and applicability of antisense-mediated knockdown for SBMA. Overall, ASO-mediated knockdown appears to be a promising strategy for the treatment of SBMA, and recent antisense-based innovations could help to boost efficacy even further.

### 2.2. Role of AR45 Isoform in Regulating Androgen Receptor Activity: Potential Therapeutic Implications for SBMA

Numerous isoforms of the AR protein are known to arise due to both internal translation initiation sites and alternative splicing, and isoform production has been shown to be linked to certain AR pathologies such as androgen insensitivity, SBMA, and prostate cancer [68,69,70]. One such example is AR-V7, a truncated AR isoform containing only exons 1–3 which is active even in the absence of androgens, and which is used as a biomarker for resistance to androgen-targeted therapy (ATT) in patients with prostate cancer [71,72,73]. In the healthy population, the most salient isoform is known as AR45, also referred to as AR isoform 2 [68]. AR45 arises from alternative splicing of a small exon called “1B” located within intron 1 approximately 22 kb downstream of the usual exon 1 [74]. The replacement of exon 1 with exon 1B leads to the production of a 45 kDa AR isoform which notably does not contain the polyglutamine motif which is expanded in cases of SBMA. 

Recently, Lim et al. found that AR45 can regulate the transcriptional activity of full-length AR by reducing the binding of AR to its transcriptional cofactor BRD4, downregulating AR-mediated transcription [74]. In the context of SBMA where mutant AR-mediated gene expression contributes to disease pathogenesis, this downregulation was theorized to have a potential therapeutic effect. To assess this, Lim et al. overexpressed AR45 using AAV9 vectors in AR100Q mice [74]. Compared to untreated controls, AR45-overexpressing mice displayed significantly increased lifespan, delayed onset of pathological weight loss, improved rotarod performance, and improved grip strength, suggesting a notable improvement in clinical phenotype following AR45 overexpression.

Given the therapeutic potential of overexpressing the AR45 isoform as identified by Lim et al., upregulating AR45 could be an interesting albeit challenging direction for future research [60,74]. One approach to achieve this could be through CRISPR Activation (CRISPRa) [75,76]. By fusing a transcriptional activator domain to a nuclease-deactivated Cas9 (dCas9) and directing it to the alternative start site with specific guide RNAs, it might be possible to elevate AR45 expression. Alternatively, ASOs could be designed to target and obstruct inhibitory elements near the alternative promoter. Additionally, small activating RNAs (saRNAs) might be utilized to enhance the transcription from the alternative promoter [77,78]. These strategies have the potential to not only amplify the AR45 levels but also concurrently reduce the amount of deleterious polyQ-AR. However, the effectiveness of these approaches remains hypothetical as experimental validation is pending, leaving its therapeutic viability for SBMA still an open question.

## 3. Conclusions

Overall, both AR knockdown and AR45 overexpression have the potential to be valuable therapeutic approaches for the treatment of SBMA. Examining the different methodologies of Lim et al. and Lieberman et al., it appears that AR knockdown results in a more robust alleviation of the SBMA phenotype than AR45 overexpression [7,74]. However, these studies were conducted with different treatment methods (AAV versus ASO), different dosages, and different SBMA mouse models, and thus direct comparisons are unreliable. As previously mentioned, the use of nanocarriers to allow blood–brain barrier penetration of ASOs has opened the door to improved treatment of diseases that affect the CNS, such as SBMA [65,66,67]. Other studies have identified protein–ASO conjugates that promote specific tissue uptake and endosomal escape of ASOs, improving their bioavailability and efficacy [79,80]. As ASO-based innovations continue to be discovered, their therapeutic potential steadily rises, and so does their potential ability to effectively treat SBMA.

## Figures and Tables

**Figure 1 genes-14-01634-f001:**
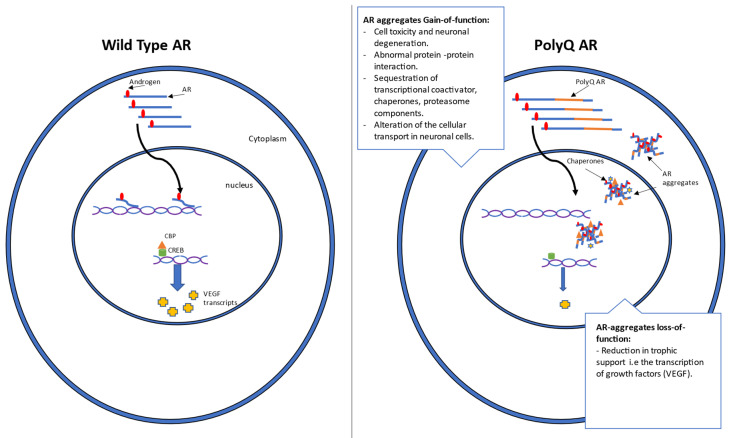
The effects of androgens binding on wild-type (WT) and polyglutamine (polyQ) androgen receptors (AR) and their function in cells. Androgen binding to AR in the cytoplasm induces a conformational change in the AR before its subsequent translocation into the nucleus, where it functions as a transcription factor. However, in the case of polyQ-AR, a distinct formation of aggregates upon androgen binding is evident. These polyQ-AR aggregates are characterized by their gain- and loss-of-function mechanisms. Some of the gain-of-function mechanisms resulting from polyQ-AR aggregates include neuronal degeneration, altered protein–protein interactions, sequestration of cellular components, and inhibition of cellular transport. On the other hand, a loss-of-function mechanism that results from polyQ-AR aggregates is the loss of AR trophic support via the downregulation of the transcription of growth factors like VEGF.

**Figure 2 genes-14-01634-f002:**
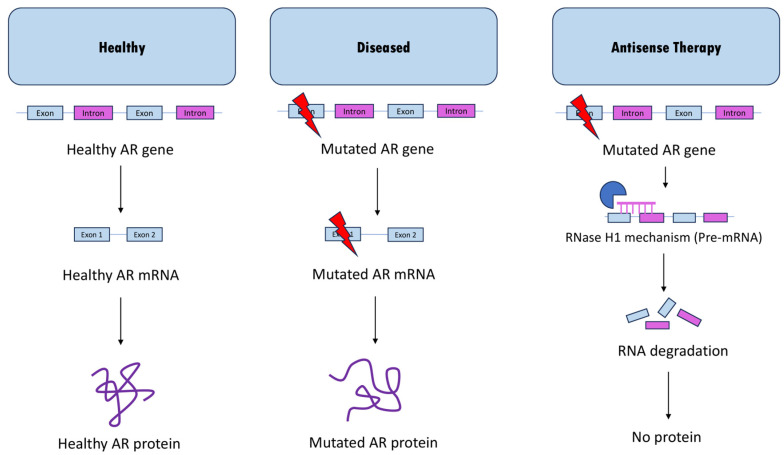
Overview of antisense therapy for SBMA. In healthy patients, the *AR* gene encodes a healthy AR mRNA, which produces a functional androgen receptor protein. In patients with SBMA, a mutation in exon 1 of the *AR* gene results in a mutated mRNA and a mutated protein. Antisense therapy developed for SBMA employs the RNase H1 mechanism, which degrades the mutated RNA.

## Data Availability

Not applicable.

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
