# Peer review of "Advancing Epidemiology and Genetic Approaches for the Treatment of Spinal and Bulbar Muscular Atrophy: Focus on Prevalence in the Indigenous Population of Western Canada"

_genes, 2023, doi:10.3390/genes14081634_

Round 1
Reviewer 1 Report
The authors make a good review on the clinical presnetation of the disease. However pathophysiology of the neuromuscular manifestations of disease should be expanded. Further in the manuscript they focus on two potential treatment strategies using ASO, which includes intratecal delivery and peripheral delivery, base don its effect onskeletal muscle. Given that SBMA is primarily a motor neuron disease with involvement of other tissues, the introduction should go though the available evidence to support primary muscle involvement. It should also explain to what extent this primary muscle pathology is expected to contribute to the phenotype, according to the available evidence.
It should also be highligther that Lieberman´s work was not designes to check the effectiveness os peripheral administration of ASO as a effective treatment for SBMA, but to demonstrate that a primary myopathy is part of the condition. Therefore, conclussions about effectiveness of this approach as a treatment would be premature, as the study was not designed for that. Furthermore, it is not correct to compare effectiveness of this approach as a potential treatment with other methods like the one descriebd my Lim et al.
Minor comments:
Filiation 4 is missing
Figure 2: Pre-m RNA should be labeled as such in figure 2 to make clear to the reader that ASO bing RNA and not DNA (as the molecule is shown to contain introns).
Author Response
Reviewer 1:
This review gives an overview on the SBMA , the molecular mechanisms being affected, and the usage of AON as potential therapeutic strategy. The paper is clearly written, well-illustrated, and covers well the current knowledge on SBMA patients.
Thank you for taking the time to review our manuscript. Your comments and thoughts are sincerely appreciated.
In the introduction section, the authors should specify that SBMA is a lower motor neuron disorder.
This has now been mentioned in line 38, and reads “Spinal and bulbar muscular atrophy (SBMA), also known as Kennedy’s disease, is an X-linked recessive neuromuscular disease with slow progression, caused by mutations in the androgen receptor gene and primarily characterized by degeneration of lower motor neurons”.
A new paragraph has also been added to mention the primary myogenic aspects of SBMA, which reads “While primarily viewed as a lower motor neuron disorder, growing evidence has identified that peripheral muscle tissues also play an important role in the SBMA phenotype. Patients with SBMA have been found to have serum creatine kinase levels consistent with a myopathic cause, and skeletal muscle satellite cells from SBMA patients have been demonstrated to have reduced proliferative ability in vitro [4–6]. Furthermore, studies treating only the peripheral muscle and not the motor neurons of SBMA mice have still displayed notable improvement in the SBMA phenotype [7]. These findings suggest that at least some of the neuromuscular symptoms of the SBMA phenotype could be caused by SBMA-related myopathy rather than the neuropathy that was previously suspected.”
In paragraph lines 98-105, it says that "Indigenous individuals with SBMA have different phenotype compared to those documented around the world." Do these patients carry a lower number of repeats on average compared to those documented around the world?
This is an excellent question given the potential links between repeat lengths and clinical phenotype, however the average number of repeat lengths in this population is consistent with the rest of the world.
Minor: ref [12] inserted twice line 85
The duplicated reference has been removed.
line 347: space missing before "However, given the therapeutic "
A space has been added.
line 352: space missing before: "One of the major advantages"
A space has been added.
Suggestion: paragraph 1.2, lines 120-132: could add a figure showing the gene protein domains and functions, conserved not conserved
Thank you for the suggestion. The authors feel that given our manuscript’s focus on antisense therapy for SBMA, a dedicated figure for the conserved protein domains between various nuclear receptors may be slightly outside the scope of this paper. For those interested, this figure can instead be found in biochemistry-focused papers such as “Structure and Function of the Nuclear Receptor Superfamily and Current Targeted Therapies of Prostate Cancer” (Porter et al., 2019) and “Androgen receptor: structure, role in prostate cancer and drug discovery” (Tan et al., 2014).
Reviewer 2 Report
This review gives an overview on the SBMA , the molecular mechanisms being affected, and the usage of AON as potential therapeutic strategy. The paper is clearly written, well-illustrated, and covers well the current knowledge on SBMA patients.
In the introduction section, the authors should specify that SBMA is a lower motor neuron disorder.
In paragraph lines 98-105, it says that "Indigenous individuals with SBMA have different phenotype compared to those documented around the world." Do these patients carry a lower number of repeats on average compared to those documented around the world?
Minor: ref [12] inserted twice line 85
line 347: space missing before "However, given the therapeutic "
line 352: space missing before: "One of the major advantages"
Suggestion: paragraph 1.2, lines 120-132: could add a figure showing the gene protein domains and functions, conserved not conserved
Author Response
Reviewer 2:
The authors make a good review on the clinical presnetation of the disease. However pathophysiology of the neuromuscular manifestations of disease should be expanded. Further in the manuscript they focus on two potential treatment strategies using ASO, which includes intratecal delivery and peripheral delivery, base don its effect onskeletal muscle. Given that SBMA is primarily a motor neuron disease with involvement of other tissues, the introduction should go though the available evidence to support primary muscle involvement. It should also explain to what extent this primary muscle pathology is expected to contribute to the phenotype, according to the available evidence.
Thank you for taking the time to review our manuscript. Your comments and thoughts are sincerely appreciated. The involvement of muscle has been better characterized, and a new paragraph in the introduction now reads: “While primarily viewed as a lower motor neuron disorder, growing evidence has identified that peripheral muscle tissues also play an important role in the SBMA phenotype. Patients with SBMA have been found to have serum creatine kinase levels consistent with a myopathic cause, and skeletal muscle satellite cells from SBMA patients have been demonstrated to have reduced proliferative ability in vitro [4–6]. Furthermore, studies treating only the peripheral muscle and not the motor neurons of SBMA mice have still displayed notable improvement in the SBMA phenotype [7]. These findings suggest that at least some of the neuromuscular symptoms of the SBMA phenotype could be caused by SBMA-related myopathy rather than the neuropathy that was previously suspected.”
It should also be highligther that Lieberman´s work was not designes to check the effectiveness os peripheral administration of ASO as a effective treatment for SBMA, but to demonstrate that a primary myopathy is part of the condition. Therefore, conclussions about effectiveness of this approach as a treatment would be premature, as the study was not designed for that.
Both the skeletal myopathy present in SBMA and the therapeutic potential of AONs for the treatment of SBMA were major foci in this paper, with the majority of their figures assessing therapeutic response. This section has been modified to better reflect this hybrid focus, and now reads at the end “This study served as preliminary justification for the therapeutic usage of AONs to treat SBMA, and also demonstrated the important role that skeletal muscle plays in the SBMA phenotype.”
Furthermore, it is not correct to compare effectiveness of this approach as a potential treatment with other methods like the one descriebd my Lim et al.
The final paragraph has been adjusted to better convey that direct comparisons between Lim et al. and the AON groups is not accurate. It now reads: “Examining the different methodologies of Lim et al. and Lieberman et al., it appears that AR knockdown results in a more robust alleviation of the SBMA phenotype than AR45 overexpression [57,76]. However, these studies were conducted with different treatment methods (AAV versus AON), different dosages, and different SBMA mouse models, and thus direct comparisons are unreliable.”
Filiation 4 is missing
We have made sure that affiliation 4 is included in the final manuscript.
Figure 2: Pre-m RNA should be labeled as such in figure 2 to make clear to the reader that ASO bing RNA and not DNA (as the molecule is shown to contain introns).
Pre-mRNA has been labelled as such to improve clarity.